# Mass Transport of Lignin in Confined Pores

**DOI:** 10.3390/polym14101993

**Published:** 2022-05-13

**Authors:** Roujin Ghaffari, Henrik Almqvist, Robin Nilsson, Gunnar Lidén, Anette Larsson

**Affiliations:** 1Department of Chemistry and Chemical Engineering, Chalmers University of Technology, SE 412 96 Gothenburg, Sweden; roujin@chalmers.se (R.G.); robnils@chalmers.se (R.N.); 2Wallenberg Wood Science Center, Chalmers University of Technology, SE 412 96 Gothenburg, Sweden; 3Department of Chemical Engineering, Lund University, SE 221 00 Lund, Sweden; henrik.v.almqvist@gmail.com (H.A.); gunnar.liden@chemeng.lth.se (G.L.); 4FibRe—Centre for Lignocellulose-Based Thermoplastics, Department of Chemistry and Chemical Engineering, Chalmers University of Technology, SE 412 96 Gothenburg, Sweden

**Keywords:** kraft lignin, delignification, mass transport, diffusion, pulping, fractionation

## Abstract

A crucial step in the chemical delignification of wood is the transport of lignin fragments into free liquor; this step is believed to be the rate-limiting step. This study has investigated the diffusion of kraft lignin molecules through model cellulose membranes of various pore sizes (1–200 nm) by diffusion cells, where the lignin molecules diffuse from donor to acceptor cells through a membrane, where diffusion rate increases by pore size. UV–vis spectra of the donor solutions showed greater absorbance at higher wavelengths (~450 nm), which was probably induced by scattering due to presence of large molecules/clusters, while acceptor samples passed through small pore membranes did not. The UV–vis spectra of acceptor solutions show a characteristic peak at around 350 nm, which corresponds to ionized conjugated molecules: indicating that a chemical fractionation has occurred. Size exclusion chromatography (SEC) showed a difference in the molecular weight (M_w_) distribution between lignin from the donor and acceptor chambers. The results show that small pore sizes enable the diffusion of small individual molecules and hinder the transport of large lignin molecules or possible lignin clusters. This study provides more detail in understanding the mass transfer events of pulping processes.

## 1. Introduction

Lignocellulosic biomass has a complex structure comprised of three main types of macromolecules: cellulose, lignin, and hemicellulose. Although various chemicals, fuels, and materials are produced from biomass, improving its utilization requires a more efficient separation of these three components. One of the main challenges is the composite structure of native biomass because it contains a variety of components with different chemical bonds, thus complicating the separation of the different species [1]. This applies not least to the removal of lignin, which is the main objective in most pulping processes. There is still a lack of fundamental understanding of how the lignin molecules are transported out from the remaining biomass matrix during delignification. This paper focuses on the transport properties of lignin in porous systems, which is relevant for pulping processes such as kraft pulping, where lignin is first dissolved and then transported out from the biomass. 

The kraft pulping process currently dominates the pulp and paper industry. It has been used for more than a century and has enabled modern pulp and paper mills to produce tailor-made fibers for various applications [2]. A modern kraft pulp mill can be designed to be self-sufficient in energy by incinerating the black liquor content (lignin, hemicelluloses, and other cooking chemicals) to generate heat and steam [3]. A clear drawback of the kraft process is its low pulp yield, which is due to the inevitable loss of carbohydrates. It is therefore desirable to increase the yield by separating the cellulose fibers and lignin more selectively whilst preserving more carbohydrates. After the main delignification reactions, liberated lignin fragments need to diffuse out toward the free liquor: the rate of delignification can therefore be controlled by either chemical kinetics of the main reactions or diffusion events during pulping [4]. So far, studies published on the kraft process have focused primarily on the diffusion of chemicals and cooking reagents into the wood chips [2,5,6,7] or on the reaction kinetics [2,4,8,9,10,11,12]. In 1999, Li et al. studied lignin diffusion in kraft delignification kinetics by investigating the effect of temperature and compared it to a calculated value based on reaction kinetics. They showed that the lignin concentration measured was much lower than the values calculated and concluded that diffusion events play an essential role in delignification processes [4]. Furthermore, Mattsson et al. suggested that, in the early part of the cook, the mass transport of OH^−^ ions into the wood chips is the rate-governing step. They showed that the main delignification reactions were completed within 10–20 min of cooking which must, however, continue substantially longer for the delignification step to be completed. This long cooking process leads to both a significant loss of carbohydrates and a lower yield of pulp. Thus, they concluded that the rate-determining step here is not the chemical kinetics of the reactions but the mass transport of soluble lignin out of the cell wall [8]. 

There are surprisingly few studies available on the mass transfer of dissolved lignin during the kraft process. Li et al. studied the intrinsic rate of diffusion of lignin molecules in fiber walls using displacement cells in alkaline conditions and analyzed the free liquor. They showed that lignin diffusion in the fiber wall is affected by the sizes of the pores and the molecules, and the electrostatic interaction between the lignin molecules and the pore walls [9]. Brännvall et al. studied the lignin content in entrapped and free liquors within kraft cooking and showed that the concentration of lignin in the liquor entrapped in the lumen is greater than that found within the fiber wall or the free liquor. They reported that the mass transfer rate of dissolved lignin is lowest from lumen liquor to free liquor [10]. Simao et al. modelled the mass transfer of dissolved lignin from the interior of the chip to bulk liquor during cooking and estimated an effective diffusion coefficient for the dissolved lignin; they also showed internal resistance to the mass transfer of dissolved lignin [5]. In a recent paper by Kawamata et al., the diffusion of lignin from wood chips during delignification using an organosolv process was shown to be the rate-limiting step. They also determined the porosity of the delignified cell walls and found that the pore size (3–10 nm) was similar to the space estimated between bundles of microfibrils in the cell wall [6]. 

The diffusion constant depends on the molecular size of the diffusing species. Lignin is not only very chemically dispersed, but it also has a broad molecular weight distribution, so determining its diffusion constant is challenging. Furthermore, the transport rate and mechanism of polymers in confinements depend on the size of the molecules relative to that of the confinements [13]. Model systems, such as side-by-side diffusion cells, have been used earlier in other research fields to refine and elucidate the effects of pore sizes on the mass transport relative to the size of diffusing molecules [14]. A diffusion cell has a donor and an acceptor chamber that are separated by a porous membrane; the mass transport rate of the molecules passing through the membrane can be calculated by analyzing samples collected from the acceptor side and knowing the concentration on the donor side. Previous studies have shown that the rate of transport was greater for membranes with larger pores [11,15,16].

Lignin self-association has been addressed in the literature before [17,18,19]. Rudatin et al. observed indications of kraft lignin association in alkalinities and stated that all molecules can participate in the association; however, to what extent they associate is different for small and large molecules. Large molecules showed substantial association between pH 12 and 13.5, while small molecules showed no association above pH 13 [17]. Norgren et al. showed—via turbidity and quasi-elastic light scattering (QELS) measurement—that kraft lignin molecules are prone to aggregate in NaOH solution in the pH range of 12 and below, and depend on the addition of NaCl (0.25–1 M) and the temperature of the solution [20,21]. Regarding the lignin transport mechanism through the fiber wall, Favis et al. showed that the size of the kraft lignin fragment can be a critical parameter [22,23]. The association in kraft lignin is strongly pH dependent [17], and can have an effect on the apparent size of lignin molecules (due to association) and consequently its transport through the membranes [9].

It is well-known that the size of the molecules affects their diffusion rate, but the molecular size relative to the size of pores can also affect the transport mechanism. For instance, in DNA electrophoresis [24,25] it has been shown that the size of the molecule relative to the size of the pore can shift the migration mechanism from being transported as coil-shaped DNA molecules to undergo reptation, whereby the DNA molecules extend in the electric field and “crawl” through the pores (like snakes moving in high grass), a transport mechanism first described by de Gennes [26]. The reptation theory was first introduced as a diffusion mechanism of linear polymers in melts or entangled polymer solutions, but has been extended to also cover reptation of branched polymers [27]. To the best of the authors’ knowledge, this transport mechanism has not been discussed so far with respect to the transport of lignin molecules.

Based on the reasoning above, model experiments using diffusion cells were therefore designed to reveal how kraft lignin molecules, with their high polydispersity and complex molecular structures, are transported in porous systems. This study presents more details pertaining to these mechanisms with the intent of improving understanding of the mass transfer events that occur in delignification processes.

## 2. Materials and Methods

### 2.1. Materials

Softwood kraft lignin (KL), extracted via LignoBoost^TM^ technology, was kindly provided by a Nordic pulp mill. The sodium hydroxide pellets used to prepare alkaline stock solutions for the dissolution of the kraft lignin were sourced from Sigma Aldrich (Taufkirchen, Germany), guaiacol, vanillin, and vanillyl alcohol standards were purchased from Merck KGaA (Darmstadt, Germany). Standard regenerated cellulose dialysis tubes (Spectrum™ Labs Spectra/Por™ with 3.5, 15, and 25 kDa molecular weight cut-off (MWCO)) and regenerated cellulose membrane filters (Whatman^®^, Dassel, Germany) with nominal pore sizes of 100 nm and 200 nm were used as membranes in the diffusion cells. 

### 2.2. Diffusion Experiments: The Set-Up of the Diffusion Cells

Mass transfer experiments were performed using tailor-made Teflon diffusion cells comprised of two half-cells. Each half-cell contained five cylindrical chambers 15 mL in volume. Regenerated cellulose membranes were placed between the chambers and covered 8 mm holes in the cells, which is where diffusion occurs. The acceptor and donor chambers were sealed with O-rings and screwed tightly together by rods to prevent leakages. NaOH solutions containing lignin were added to the donor chamber, whilst blank solutions of the same NaOH concentration were added to the acceptor chambers. The diffusion cells were placed on an orbital shaker (PSU-20i, LabTeamet, Grant Instruments, Cambridge, UK) operating at 100 rpm for the entire duration of the experiment. 

### 2.3. Diffusion Experiments: The Procedure for Lignin Diffusion

The donor solution was prepared by dissolving 100 mg of kraft lignin in 250 mL of 0.1 M NaOH solution (pH of the solution after dissolution of lignin was about 12.9). The solutions were stirred for at least 2 h to ensure complete dissolution and to obtain clear solutions. Prior to starting the experiments, regenerated cellulose (RC) membranes with 3.5, 15, and 25 kDa MWCO (or nominal pore size corresponding to 1–2, 3–4, and 4–5 nm, respectively according to the supplier), and the membranes with a nominal pore size of 100 and 200 nm were all pretreated in a 0.1 M NaOH solution for at least one hour before being mounted into diffusion cells. Measures of 15 mL of donor solution and 15 mL of blank solution were poured simultaneously into the respective chambers, the set-up of which is described in Section 2.2 above. The diffusion chambers were kept on an orbital shaker (100 rpm) at room temperature for the duration of the experiments. Then, 1 mL samples were drawn from the acceptor chamber at 24 h intervals, and replaced with fresh 0.1 M NaOH solution, for a total of seven days. The samples withdrawn were analyzed spectroscopically by UV absorbance spectroscopy using a Cary60 UV–vis spectrophotometer (Agilent Technologies, Inc., Santa Clara, CA, USA) using Cary WinUV scan application software (Version 5.0.0.999, Agilent Technologies, Inc., Santa Clara, CA, USA) to record the spectra. The samples were scanned from 600 nm to 200 nm at the rate of 4800 nm/min, and their lignin concentration was quantified by sample absorbance at 290–300 nm based on a calibration curve of lignin samples in the concentration range of 10 to 40 mg/L. The dilution of acceptor solution upon sampling is accounted for in the concentration calculations. The diffusion experiments were conducted in triplicate. 

### 2.4. Size-Exclusion Chromatography

Samples were collected from the diffusion cells at varying time intervals and the size of the lignin molecules was analyzed by a size exclusion chromatography (SEC) system (HPLC Pump 1515, Autosampler 717plus, Waters, Milford, MA, USA). Two analytical columns packed with Superdex 200 Increase (300 × 10 mm, 9 μm) and Superdex 30 Increase (300 × 10 mm, 9 μm) (Cytiva, Uppsala, Sweden) were used for separation. The columns were operated at ambient temperature and eluted with 0.1 M NaOH solution (analytical grade) as the mobile phase at a flowrate of 0.5 mL/min. Polyethylene glycol (PEG) standards ranging from 200 to 35,000 Dalton (Merck Schuchardt OHG, Hohenbrunn, Germany) were used for calibration. The system was equipped with a Waters 2414 refractive index (RI) detector and a Waters 2487 dual-wavelength UV detector, with absorbances at 280 and 350 nm being used for detection. The RI detector was used for calibration using the PEG standards, whereas—for detection of lignin molecules—the UV detector responses at 280 nm and 350 nm were recorded to monitor the structural changes in the lignin molecules. The delay between UV and RI signals was only a few seconds, which is minimal compared to the analysis time which is about 120 min.

### 2.5. Ultra-High Performance Liquid Chromatography (UHPLC)

Samples taken from the acceptor side of the diffusion cells were first neutralized with HCl and then analyzed for vanillin, vanillic acid, and guaiacol in a Waters H-Class UPLC system using a method described in the literature [28]. A gradient method with water (A) and acetonitrile (B), both with 10 mM formic acid, was used to separate the lignin monomers in an Agilent InfinityLab Poroshell 120 EC-C18 column (100 × 4.6 mm, 4 μm). The flow rate was 1.0 mL/min and the gradient for acetonitrile (B) was as follows: 0 min 3%, 12 min 15%, 15 min 15%, 20 min 80%, 20.1 min 90%, 25 min 90%, 25.1 min 3%, and 35 min 3%. The injection volume was 3 µL and the column temperature 50 °C. Detection was performed using a PDA detector at 280 nm with baseline correction at 350–400 nm.

### 2.6. UV–Vis Absorbance Measurements 

Changes in the absorption characteristics of lignin with pH were studied spectroscopically using the same spectrophotometer as above. Samples were collected from the acceptor side and neutralized to pH 7 by adding HCl. UV–vis spectra were recorded before and after neutralization.

## 3. Results

### 3.1. Transport through Membranes in Diffusion Cells

The rates of transport of the lignin molecules were determined using diffusion cells. Regenerated cellulose membranes of various pore sizes were exposed to lignin in 0.1 M NaOH solutions on the donor side. The lignin molecules passed through the pores in the membranes into the acceptor solution, where their concentrations were determined by UV–vis absorption. The concentration of lignin on the acceptor side shows a linear increase with time (Figure 1). The curve does not exhibit any abrupt changes with time, indicating that the diffusion process in the membrane follows a steady trend.

As expected, there is an increase in the diffusion rate with increasing pore size. The transport rates of lignin samples passing through the membranes with the largest pore sizes (RC100 and RC200) were about 10 times greater than for the smallest. The diffusivity calculated for lignin over RC200 and RC3.5 were 39 × 10^−14^ and 3.8 × 10^−14^ m^2^ s^−1^, respectively (see Appendix A). The reduced transport rate can be explained by the fact that large lignin molecules (with a larger hydrodynamic radius) and/or lignin clusters are maybe unable to enter and pass through the pores of smaller size, whilst small molecules can. This was verified by using SEC to determine the molecular size distribution (Figure 2).

Determining the absolute molecular weights of lignin molecules is challenging, so the focus of the discussion here is placed on the retention times, R_T_, in the chromatograms. The Appendix A section shows, as a matter of interest, that the average molecular weight of lignin is measured as being 3.9 kDa, with a polydispersity index of 2.8. This agrees with values reported earlier for kraft lignin [28]. 

The SEC chromatogram in Figure 2a shows that intensity increased as the size of the pores in the membranes increased, thereby directly confirming the diffusivity data in Appendix A and the observations in Figure 1. The intensities also increased with pore sizes for retention times, R_T_, up to 73 min, where the sizes according to the calibration curve correspond to around 0.5 kDa (pink arrows in Figure 2a,b). The SEC chromatograms also show that, with increasing pore size, the chromatography peaks increase in intensity at smaller R_T_, i.e., for lignin molecules with a greater hydrodynamic radius, R_h_ (e.g., the blue arrow in Figure 2c). It can also be observed that the intensities in the SEC chromatogram for the RC25 and RC100 samples collected after different times increased both with time and pore size of the membranes (Figure 2c,d), thereby resembling the results in Figure 1. When comparing the chromatograms (both 280 and 350 nm) in Figure 2, it is evident that there is a significant difference between the samples collected from the acceptor cells. At early retention times (~60 min), a clearly visible peak is observed for RC100, RC200, and donor spectra, whereas RC3.5, RC15, and RC25 do not show any intensity at these retention times. This indicates that large molecules have difficulty in passing through small pores, like those observed initially in the fiber walls during the pulping process, thus slowing down the delignification process significantly. Small pore sizes give rise to size fractionation of the lignin sample: membranes with smaller pores hinder the transport of lignin more than those with larger pores. However, the increase in intensity in the SEC chromatogram for RC25 starts at approximately when R_T_ is equal to 58 min, which should correspond to a molecular weight of 4 kDa according to the calibration. Even though the calibration curve might not be perfectly accurate for lignin, it is remarkable that a large portion of the lignin molecules do not pass the RC25 membrane. 

Furthermore, the SEC analyses were made using the detectors at 280 and 350 nm in Figure 2a and Figure 2b, respectively. The chromatograms are not identical, indicating that the lignin molecules have different chemical compositions at different retention times; the differences are more pronounced at longer R_T_, i.e., small R_h_. This was investigated further by comparing the UV–vis spectra of the samples in the acceptor chamber to those of the donor solution, see Figure 3a. Here, all data were scaled to show the same absorbance value at 300 nm. The UV–vis spectra of samples collected from the acceptor side after passing membranes with small pore sizes (RC3.5, RC15, and RC25) differ clearly from those collected from the donor side: an additional distinct peak at around 350 nm in wavelength decreases with increasing pore size. The difference in the spectra of the acceptor and donor sides indicates that a change has occurred in the chemical structure of the lignin transported across the membrane.

Interestingly, the UV–vis spectra in Figure 3a show a quasi-isosbestic point around 390 nm, suggesting a prevalence of lignin molecules in two different molecular forms. The green arrow in Figure 3a at the 420 nm wavelength shows an increase in absorbance with increasing pore size, indicating that the samples with increasing pore sizes contain species that scatter light.

Absorbance spectra of the different samples collected from the acceptor at 165 h were recorded. To investigate the effect of pH on absorbance, hydrochloric acid was added to the samples and the absorbance recorded again. Absorbance spectra changed by the decrease in pH (Figure 3b), where the clear peak at 350 nm for RC 3.5, RC15, and RC25, as well as the weak shoulder for RC100 and 200, disappeared (indicated by the black arrow in Figure 3b). A peak at 280 nm, with a weak shoulder at 320 nm, appeared when the pH was decreased. Thus, it can be concluded that the chemical composition of the solutions collected on the acceptor side in the diffusion cell differs, to some extent, from the donor side. Which suggests a possible change in chemical composition of acceptor samples depending on the pore size of the membrane in use.

### 3.2. Properties of the Lignin Transported through Membranes in Diffusion Cells

So far, the following have been observed:The amount of lignin transported from the donor side to the acceptor side increased with time as the pore size of the intermediate membrane increases.The absorbance intensity in the SEC increased with time and increasing pore size, even for small fragments.The intensity in the SEC chromatogram started to increase at lignin sizes that correspond to a factor of almost 10 times smaller than the nominal pore sizes.Corresponding SEC curves recorded at 280 and 350 nm were not identical, especially for the smaller lignin molecules.UV–vis absorbance spectra from the acceptor cells of the different membranes were not identical.

The question is: how should these results be interpreted? Inspired by the literature, the hypothesis we put forward here is that lignin molecules can associate to form clusters of lignin, and that these clusters can be transported from the donor to the acceptor side through large pores but not the small pores. Here, we refer to kraft lignin clusters as a dynamic self-assembled association of kraft lignin molecules which exhibit a micellar-like behavior in a way that molecules can join or leave the clusters in a dynamically. Furthermore, the ability of lignin molecules to participate at a specific pH in a cluster depends on their chemical properties and size. Micellization has a huge impact on the net transport rate, since larger-sized micelles are transported much more slowly [31,32] and it is likely that self-assembled associations could influence the transport rate in a similar way.

#### 3.2.1. Chemical Differences of the Lignin Transported through Membranes in Diffusion Cells

The ionization of the hydroxyl groups of lignin in alkaline conditions has previously been shown to induce both bathochromic and hyperchromic shifts in UV–vis spectra [33]. It has also been shown that phenolic lignin units conjugated in the αα-position have an extra absorbance maximum around 350 nm in alkaline solutions. When the pH is reduced to neutral conditions, however, only one maximum absorbance remains: around 280–300 nm and with decreased intensity. At the same time, a maximum at about 300 nm in alkaline conditions indicates the presence of unconjugated phenolic lignin units, and this peak is shifted to 280 nm when the pH is reduced to neutral conditions [33,34,35]. Returning to Figure 3a,b, it is likely that the clear peak at 350 nm in wavelength for RC3.5 is due to conjugated lignin units: the relative number of conjugated to unconjugated units decreases as the pore size increases. The graphs of pH changes, shown in Figure 3b, are in accordance with the lignin becoming more ionized at a higher pH and therefore they show an extra absorbance at 350 nm in the case where charged conjugated bonds are present. This indicates that the majority of the lignin molecules entering the acceptor chamber of the small pore membranes are probably highly charged conjugated lignin.

Zakis et al. [29] and Gärtner et al. [30] have developed a method for comparing amounts of conjugated (carbonyl- or double-bond-conjugated phenols such as stilbenes and quinones [36]) and unconjugated lignin, which is applied here to the different acceptor samples. They calculated Δε by subtracting the absorbance spectra in alkaline and neutral conditions (Figure 3c). Even though this method has been shown to underestimate the phenolic hydroxyl content available in kraft lignin samples, it can be helpful for comparison purposes [37]. Figure 3c shows Δε for the samples collected at the acceptor side after passing different membranes and Figure 3d shows Δε at 350 nm relative to Δε at 300 nm, which correlates linearly to the ratio of conjugated phenols to unconjugated phenols [30,34,38]. The RC3.5, RC15, and RC25 samples have a higher ratio than the RC100 and RC200 samples: this indicates that a higher fraction of conjugated phenolic structures relative to unconjugated has been transported to the acceptor side through the membranes of smaller pore size, thereby confirming that a chemical fractionation occurs in the diffusion cell experiments. The absorbance spectra were measured for solutions in which the pH had been lowered and then restored to high levels; after this pH treatment, the spectra displayed the peak at 350 nm again, showing that ionization is reversible. 

The presence of more conjugated structures in the solution obtained from samples from the acceptor side can be due to various reasons: one can be that the transported molecules are smaller fragments of lignin, such as monomers and oligomers, that were formed mainly during the kraft process when aryl-ether bonds were broken. The breakage of these bonds occurs through quinone methide intermediates and/or the formation of conjugated bonds on α or β carbons and results in the formation of conjugated phenolic end groups, which can easily be charged in alkaline solutions [2,39]. It is therefore essential to compare the sizes of the lignin species and the concentration of the monomers that passed through the various membranes, both of which comparisons were carried out using chromatography techniques.

#### 3.2.2. Size-Fraction of Lignin Transported through Membranes in Diffusion Cells

The ionized and highly charged lignin molecules are less likely to associate [20], which is indicated by the lesser light scattering at higher wavelengths for the acceptor samples of RC3.5, RC15, and RC25 than those of RC100, RC200, and the donor. Such associations of lignin, as suggested by Norgren et al. [20,21], are likely to affect the transport mechanism in the diffusion cells. The first argument supporting the fact that lignin association occurs in the diffusion cells and alters its transport through a porous membrane is the limited transport observed in the case of the RC25 membrane despite its molecular weight cut-off of 25 kDa, and that kraft lignin with a M_w_ of an average around 4 kDa was used. The pores in this membrane should therefore be sufficiently large to allow the majority of lignin molecules to pass without any particular hindrance. Nevertheless, Figure 2c,d show that the fraction of large molecules that passed the RC25 membrane is lower than for RC100 and probably indicates the formation of clusters [40]. Most of these possible clusters can however pass through the large pores in the RC100 and RC200 membranes. At the same time, molecules that associate to clusters may be able to pass the RC25 membrane if the clusters dissociate to individual molecules or smaller clusters and these pass the membrane.

### 3.3. A Critical Lignin Size Controls the Transport Mechanism

The transport mechanism for a lignin molecule with a specific R_h_ was investigated by making the following derivations. Let us consider steady-state diffusion between two adjacent cells through a thin membrane, where transport resistance in the stagnant layers surrounding the membrane is negligible and the concentrations at the membrane boundaries are the same as in the bulk in the donor and acceptor chambers, respectively. Using Fick’s law and having the volumes in the donor and acceptor chambers equal, the diffusion of lignin through the model membranes can be described using the equation [41]
(1)$$\left(\frac{2\;D_{i}\;S}{V\;h}\right)\;t=-Log\left(\frac{C_{di}-2C_{ai}}{C_{di}}\right)$$
where *D_i_* (m^2^/s) is the diffusion coefficient of specie *i*, *S* (m^2^) is the surface area, *V* (m^3^) is the volume of the donor and acceptor chambers, *h* (m) is the thickness of the membrane, and *t* (s) is time. *C_di_* and *C_ai_* are the concentrations of species *i* in the donor at the start and in the acceptor at time t [41]. Based on Equation (1), *D_i_* is proportional to
(2)$$D_{i}\;\propto -Log\left(1-\frac{2C_{ai}}{C_{di}}\right)$$

*C_a_* is much smaller than *C_d_* and, using the Taylor series on $-Log\left(1-\frac{2C_{ai}}{C_{di}}\right)$ around 0, expression (2) can be rewritten as
(3)$$D_{i}\;\propto \;\frac{C_{ai}}{C_{di}}$$

According to Stokes–Einstein equation, *D_i_* is inversely proportional to the hydrodynamic radius and therefore
(4)$$D_{i}\propto R_{hi}^{-1}\;$$

Depending on the configuration of the molecule in the solution, its hydrodynamic radius changes in relation to its molecular weight. A *φ* parameter was therefore defined that would indicate how the *M_w_* of the molecules has affected their diffusion within different pores. In the SEC system, it is known that retention time has a linear relation to $Log_{10}\;\left(M_{w}\right)$, therefore
(5)$$R_{hi}\;\propto \;M_{wi}^{\varphi }\;\propto {\left(10^{-R_{Ti}}\right)}^{\varphi }$$

From expressions (3) to (5), it follows that
(6)$$Log_{10}\;\left(\frac{C_{ai}}{C_{di}}\right)\propto \;\varphi R_{Ti}\;$$

The way in which the diffusion mechanism of different sized molecules is affected by their molecular weight was investigated by plotting $Log_{10}\;\left(\frac{C_{a}}{C_{d}}\right)$ against R_T_ for all membranes, as seen in Figure 4. If the lignin molecules migrate through all of the pores via a similar mechanism, the graphs should follow the same slope for all R_h_ and thus also R_T_, without any abrupt changes. A linear dependence can be seen in the figure for RC100 and RC200 between 55 and 80 min. Between 48 and 55 min, there is a somewhat steeper slope. Similarly, parallel lines at long R_T_ are displayed for RC25, RC15, and RC3.5, implying a small dependence of D on R_T_. However, a critical retention time, R_TC_—defined as the intercept between the slopes—occurs at around 65, 66, and 69 min for RC25, RC15, and RC3.5, respectively. These R_TC_ values correspond to the critical M_w_ and M_wc_, which are 2.0, 1.8, and 1.0 kDa, for RC25, RC15, and RC3.5, respectively. The M_wc_ calculated here is about 3–12 times smaller than the MWCO reported for the corresponding membranes. This opens up for the interpretation that the transport mechanism in the diffusion cells changes for lignin molecules above or below the R_TC_, which may be explained by lignin molecules associating with clusters, and when the size of the clusters becomes too large relative to the pore size, the transport mechanism for the lignin is altered and slowed down. The UV–vis curves in Figure 3a support the occurrence of clusters in RC100 and RC200 by these showing more scattering at higher wavelengths than the RC3.5, RC15 and RC25 membranes. The different transport mechanism could also be due to the clusters and/or individual lignin molecules changing in configuration to ease their transport through the pores, as discussed below.

Eventually, the clusters that are transported to the acceptor side through the large pore membranes may partially dissociate due to their formation being concentration-dependent [19], because clusters that have reached the acceptor chamber are exposed to a lignin concentration of less than 1/10 of the concentration in the donor chamber, therefore enabling dissociation. The dissociation process can explain why the intensity increases with increasing pore size at long R_T_.

UHPLC analyses were carried out for the standard monomers present in lignin (vanillic acid, vanillin, and guaiacol) on samples collected from the acceptor side of the diffusion cells after 168 h (Figure 5). The very similar concentrations of these molecules after passing different pore sizes seemingly contradicts the involvement of small units in the clusters, and possibly also that the transport of clusters to the acceptor side is the reason for the different concentrations in the acceptor side observed at, e.g., R_T_ equal to 73 min in the SEC curves. However, as the small, charged molecules in Figure 5 have a molecular weight <200 g/mol, they may be transported with the solvent and appear in the solvent peak at 85 min in the SEC graphs, which is not in the region of the R_T_s that was compared earlier.

This section may be divided by subheadings. It should provide a concise and precise description of the experimental results, their interpretation, as well as the experimental conclusions that can be drawn.

## 4. Discussion

### 4.1. Transport Mechanism of Lignin in Porous Environments

Figure 6 is a schematic summary of the diffusion process of lignin over the membranes: Figure 6a illustrates how molecules and dynamic clusters of lignin can pass through large pores, while Figure 6b shows how some large lignin molecules and clusters are hindered by small pores.

The transport of clusters is described in Figure 6c. The association of lignin into clusters is similar to that of surfactants associating into micelles. It has been shown that the transport of surfactants over porous membranes above their critical micelle concentration (CMC) is independent of the total concentration of the surfactants on the donor side [32], which is in direct contrast to the transport of non-associating molecules. This is because the transport rate of micelles is very low, and only the single surfactants (unimers) are initially transported successfully through such membranes. Nagata et al. showed that the pore size influences the transport of micelles in a membrane, and that when the radius ratio between micelle and pore exceeded 0.05, this migration was disturbed [42]. If the transport of lignin clusters is assumed to be similar to that of micelles in pores, with a similar radius ratio between the penetrant and the pore, it is not surprising that the rate of transport of lignin through RC25 membranes is zero at a radius ratio of about 1/10 (4 kDa relative 25 kDa, see above). Moreover, the fact that the transport rate is the same for RC100 and RC200 indicates that the sizes of the clusters are below the critical cluster-to-pore sizes.

An alternative mechanism for the change in transport mode is since the molecular shape of lignin molecules changes when their size is close to that of the pores (Figure 6d). Whilst the 3D shape of lignin molecules is under debate, it has been suggested by Crestini et al. that larger softwood kraft lignin molecules (acetone insoluble fraction) have a “less branched, more polymer like” structure [39]. Assuming that the hydrodynamic radii of lignin molecules are comparable to the pore size of membranes, it may be speculated that the lignin molecules are able to rearrange their molecular shape before entering the pores, and eventually start to migrate via a reptation mechanism, see Figure 6d. Polymers—whether linear, branched, or closed circular—can migrate in narrow channels using the reptation mechanism [27] whereby the molecules crawl (like a snail) through the constraining network [26]; a reptation-like mechanism has already been found for the diffusion of other polymers in confined systems. Such rearrangement of lignin would allow diffusion of large molecules through small pores (e.g., RC3.5, RC15, and RC25) but at a low rate; meanwhile, for larger pores (RC100 and RC200), most could keep their molecular shapes and pass through the pores without any rearrangement being necessary (Figure 6a). The reptation mechanism can be a possible mechanism for transporting lignin molecules through small pores even if it requires the lignin molecules to be dynamic enough to adopt extended shapes, which may be questionable.

### 4.2. Effects on the Delignification Process

Mass transfer in the pulping process occurs mainly through the fiber cell wall, lumen, and pits: the lumina are the hollow cavity in the middle of fibers; and holes, also known as pits, enable communication between adjacent lumina. Pits can be either simple or bordered: the cavity width is almost constant in the former and narrows down toward the lumen in the latter. Both types of pits have a membrane comprised of pores that are approximately a couple of nanometers in size [43]. The fiber wall is built up of cellulose micro and macrofibrils, and the distance between them is similar to the size of the pits [6]. The pores in the RC3.5, RC15, and RC25 membranes used here resemble the pores in the pit membranes in wood that limit mass transport significantly. When the diameter of the pores here is comparable to the size of lignin molecules in the solution, the molecules are more confined within the pores, which hinders their diffusion. Since both lignin molecules and pores carry various functional groups that can be charged at high pH levels, electrostatic interactions between pore walls of the membrane and lignin molecules are possible. Once soluble lignin has passed through a fiber wall into the lumen, it is transferred into the bulk liquor via large cell-wall pits. In the experiments carried out here, membranes with 100 nm and 200 nm nominal size pores (RC100 and RC200, respectively) represent these cell-wall pits. The diffusion rate is almost similar in both the RC100 and RC200 membranes because the diameters of the pores are much larger than the lignin molecules, or clusters, passing through them, and therefore the confinement of molecules within the pores does not limit diffusion.

This study suggests that lignin molecules can be partially associated into clusters, and that the transport rate of diffusion of these clusters is both slow and strongly dependent on pore size. This indicates that, in the delignification process, the number of pores and their size are initially small, which limits the amounts of lignin that can be transported out from the cell wall. Lignin can be transported more rapidly as the pores enlarge over time as pulping progresses. However, the critical issue is not only the size of the lignin molecules. It may also be that the size of the associated clusters and pores. This means that the average size of the pores that limit such transport should be increased in order to achieve higher efficiency in delignification processes.

## 5. Conclusions

The diffusion rates of lignin molecules within various pore sizes of cellulose membranes were compared using diffusion cells. The main advantage of this method is that external mass transfer resistance and delay times can be avoided while a wide range of experimental conditions can be applied to the system, such as various alkalinities, temperatures, and membranes of different types, as well as pore size. The diffusion rates of lignin molecules were found to increase as the pore size increased, when the pores were in the nanometer range (RC3.5, RC15, and RC25). It was found that pores in the range of 10 nm or below allowed only lignin molecules of lower M_w_ to pass, while larger pore membranes (RC100 and RC200) let most of the lignin molecules/clusters through.

The UV–vis spectra of samples taken from the acceptor side differ from those taken from the donor side. In addition to the conventional 300 nm UV–vis peak for lignin in high alkalinity, a distinct peak around 350 nm can be observed in the acceptor spectra, which can be correlated to lignin molecules with conjugated bonds in the aliphatic region. An interpretation of this can be that the relative concentration of conjugated to unconjugated phenolic structures is higher in the acceptor than in the donor after passing through small pores. The SEC studies showed that the diffusion of many large lignin molecules was hindered significantly when the pores were small. SEC data showed a critical molecular size, where a change in the diffusion mechanism was observed. Small, charged lignin molecules similar in size to the pores can pass through as single molecules, whereas large lignin molecules are more involved in clusters and need to dissociate to pass small pores or potentially crawl through them via reptation. When clusters in the donor solution are larger than the pores, their transport becomes limited, and they must dissociate before they can pass through. Lastly, the possible interactions between the cellulosic membrane and the dissolved lignin molecules should not be forgotten. In the highly alkaline conditions per, of these experiments, charges are available on both molecules and the membranes and this can affect their diffusion process. This study provides more detail of the diffusion of lignin molecules in confinement in accordance with the size of the confinements; further studies using the present method are recommended, especially with respect to the effects of pH and temperature.

## Figures and Tables

**Figure 1 polymers-14-01993-f001:**
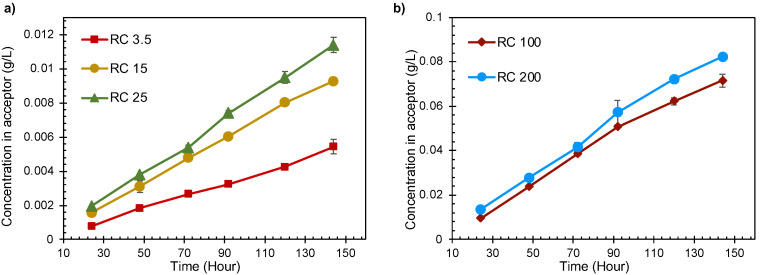
Concentration of lignin in the acceptor chamber versus time through different membranes. (**a**) RC3.5, RC15 and RC25; (**b**) RC100 and RC200.

**Figure 2 polymers-14-01993-f002:**
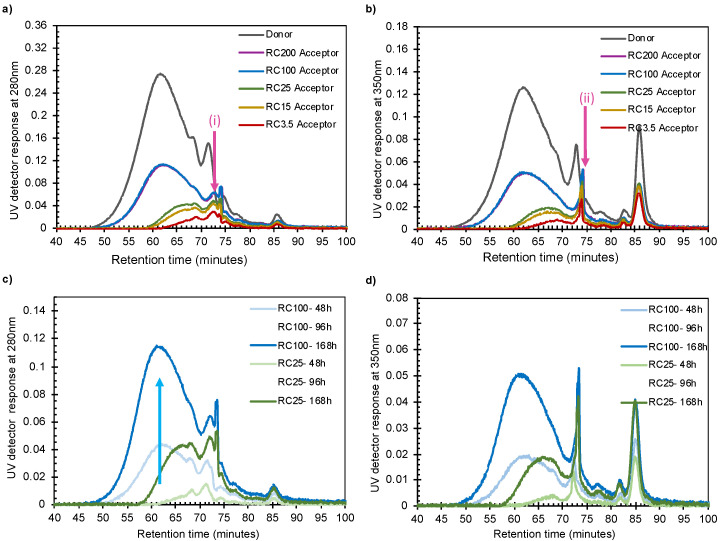
UV detector response of SEC versus retention time at (**a**) 280 nm and (**b**) 350 nm. Changes in UV detector response at (**c**) 280 nm and (**d**) 350 nm for three diffusion times with RC100 and RC25 membranes. (i) Refers to the peak at 72 min of the 280 nm chromatogram, which is missing in the 350 nm chromatogram. (ii) Refers to peak at 73 min, which is larger relative to the earlier peaks in the 350 nm chromatogram but not as large when compared to the earlier peaks.

**Figure 3 polymers-14-01993-f003:**
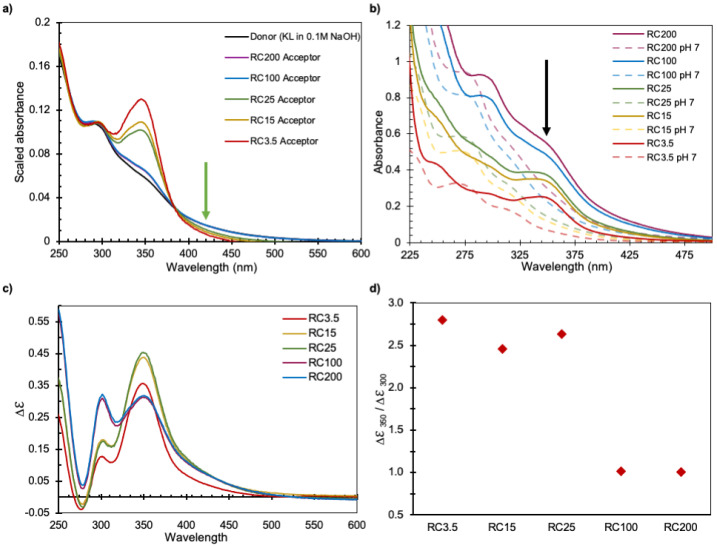
(**a**) Scaled UV-vis spectra of samples taken from the acceptor side of diffusion cells after 168 h; (**b**) UV-vis spectra of original samples taken from the acceptor side of diffusion cells (in 0.1 M NaOH), and their respective neutralized spectra; (**c**) Ionization difference spectra (Δε) of samples from acceptor chambers collected at 164 h; (**d**) Ratio of ionization difference (Δε) at 350 nm to 300 nm, which is proportional to the relative concentration of conjugated to unconjugated units [29,30].

**Figure 4 polymers-14-01993-f004:**
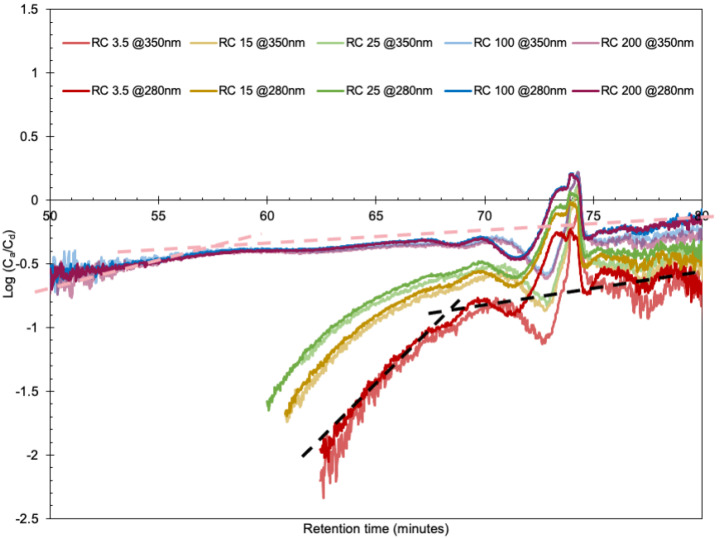
Log(C_a_/C_d_) versus retention time. The dashed lines show the point at which the diffusion mechanism changed in the RC3.5 and RC100 membranes.

**Figure 5 polymers-14-01993-f005:**
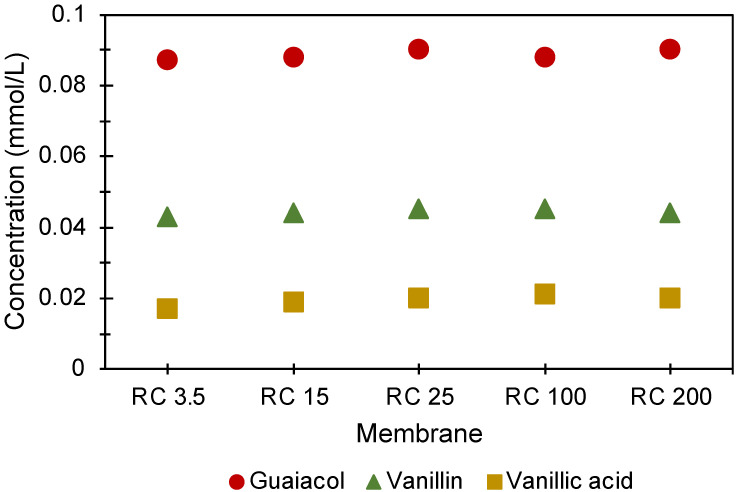
Concentration of various monomers determined from the UHPLC results obtained on the acceptor side of five different membranes.

**Figure 6 polymers-14-01993-f006:**
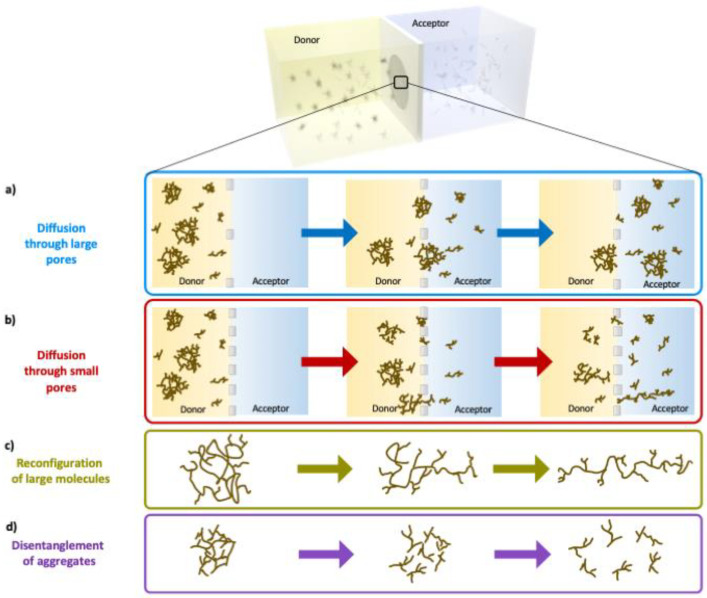
Schematic diagram of the diffusion of lignin molecules through membranes in diffusion cells (**a**,**b**), disentanglement/dissociation of lignin clusters that allows them to pass through a membrane (**c**), and reconfiguration of large lignin molecules that enables them to pass through a membrane via a reptation-like mechanism (**d**).

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
