# Peer review of "Mass Transport of Lignin in Confined Pores"

_polymers, 2022, doi:10.3390/polym14101993_

Round 1

Reviewer 1 Report

Dear authors,

I have read and analyzed your paper and made the following comments/sugestions:

  1. Please update the statement regarding the chemical composition of black liquor (line 45);
  2. please adequately describe the procedure for GPC(SEC) system calibration. I have made my comments on the pdf version - lines 178-181. ...SEC-GPC calibration by using PEG is appropriate for the determination of the molecular weight distribution curves of polysaccharides and therefore the usage of RI detector seems appropriate. On the other hand, since the detection of lignin is made by UV, it would have been much more better to use standards as Poly(styrenesulfonic acid sodium salt ( an example is CAS number 9080-79-9 supelco). Please explain and justify the need of the two detector system and configuration order. Please add  RI signal chromatograms to your supplementary material as support - (superpose chromatograms of standards if you consider necessary or superpose UV/Refractive Index chromatograms).
  3. .. there is a reference missing at line 209,240,268 and so on.
  4. what do you mean by conjugated or unconjugated lignin (structures)? maybe some representation would help...please add them to supplementary.
  5. most equations seem to miss some symbols for multiplying ...(X) divisions and =...is there anything wrong with the math editor... it looks confusing. please check pdf version.
  6. In line 582 concerning supplementary information there is a video mentioned... have you forgotten to upload it? I could not find it on the MDPI site.

Author Response

Dear Reviewer 1,

Thank you for your kind comments. We have considered your comments carefully and improved the manuscript accordingly. Please see the attached file for our responses to the questions.

We hope that our responses full fills your requirements.

Best regards,

Anette Larsson
Roujin Ghaffari

Reviewer 2 Report

            The authors did a systematic study of the mass transport of lignin in confined pores using size exclusion chromatography and UV-Vis spectroscopy. The experimental evidence supports the conclusion that the small pore sizes enable the diffusion of small individual molecules and hinder the transport of large lignin molecules or possible lignin clusters. I would recommend the acceptance of the manuscript in its current format.

Author Response

Dear Reviewer 2,

Thank you for your kind comments. We are happy that our manuscript has met your standards.

Sincerely,

Anette Larsson,

Roujin Ghaffari